# Megalencephalic Leukoencephalopathy with Subcortical Cysts Disease-Linked MLC1 Protein Favors Gap-Junction Intercellular Communication by Regulating Connexin 43 Trafficking in Astrocytes

**DOI:** 10.3390/cells9061425

**Published:** 2020-06-08

**Authors:** Angela Lanciotti, Maria Stefania Brignone, Marcello Belfiore, Sandra Columba-Cabezas, Cinzia Mallozzi, Olimpia Vincentini, Paola Molinari, Tamara Corinna Petrucci, Sergio Visentin, Elena Ambrosini

**Affiliations:** 1Department of Neuroscience, Istituto Superiore di Sanità, Viale Regina Elena 299, 00161 Rome, Italy; angela.lanciotti@iss.it (A.L.); mariastefania.brignone@iss.it (M.S.B.); sandra.columbacabezas@iss.it (S.C.-C.); cinzia.mallozzi@iss.it (C.M.); tcpetrucci.petrucci61@gmail.com (T.C.P.); 2National Centre for the Control and Evaluation of Medicines (CNCF), Istituto Superiore di Sanità, Viale Regina Elena 299, 00161 Rome, Italy; marcello.belfiore@iss.it; 3Department of Food Safety, Nutrition, and Veterinary Public Health, Istituto Superiore di Sanità, Viale Regina Elena 299, 00161 Rome, Italy; olimpia.vincentini@iss.it; 4National Centre for Drug Research and Evaluation, Istituto Superiore di Sanità, Viale Regina Elena 299, 00161 Rome, Italy; paola.molinari@iss.it (P.M.); sergio.visentin@iss.it (S.V.)

**Keywords:** astrocyte, connexin, gap junctions, physiology, physiopathology, ERK1/2, leukodystrophy, rare diseases, blood–brain barrier, GlialCAM

## Abstract

Astrocytes, the most numerous cells of the central nervous system, exert critical functions for brain homeostasis. To this purpose, astrocytes generate a highly interconnected intercellular network allowing rapid exchange of ions and metabolites through gap junctions, adjoined channels composed of hexamers of connexin (Cx) proteins, mainly Cx43. Functional alterations of Cxs and gap junctions have been observed in several neuroinflammatory/neurodegenerative diseases. In the rare leukodystrophy megalencephalic leukoencephalopathy with subcortical cysts (MLC), astrocytes show defective control of ion/fluid exchanges causing brain edema, fluid cysts, and astrocyte/myelin vacuolation. MLC is caused by mutations in MLC1, an astrocyte-specific protein of elusive function, and in GlialCAM, a MLC1 chaperon. Both proteins are highly expressed at perivascular astrocyte end-feet and astrocyte-astrocyte contacts where they interact with zonula occludens-1 (ZO-1) and Cx43 junctional proteins. To investigate the possible role of Cx43 in MLC pathogenesis, we studied Cx43 properties in astrocytoma cells overexpressing wild type (WT) MLC1 or MLC1 carrying pathological mutations. Using biochemical and electrophysiological techniques, we found that WT, but not mutated, MLC1 expression favors intercellular communication by inhibiting extracellular-signal-regulated kinase 1/2 (ERK1/2)-mediated Cx43 phosphorylation and increasing Cx43 gap-junction stability. These data indicate MLC1 regulation of Cx43 in astrocytes and Cx43 involvement in MLC pathogenesis, suggesting potential target pathways for therapeutic interventions.

## 1. Introduction

Astrocytes, the most abundant cells of the central nervous system (CNS), by bridging synapses and blood vessels, perform essential functions for the maintenance of brain homeostasis and neuronal and synaptic activity. To accomplish these tasks, astrocytes arrange themselves as a highly interconnected network of cells able to rapidly exchange ions, small molecules, and metabolites between them [1,2,3]. The intercellular connectivity is ensured by the presence of gap junctions. These are connected hemichannels located on the opposite cellular membranes and composed of six subunits of connexin (Cx) proteins, including Cx43, the dominant Cx expressed in astrocytes (see References [1,4,5,6] and the references therein). Gap-junction coupling between astrocytes allows the control and coordination of the neuronal activity and brain energy metabolism through the delivery of nutrients to neurons, the spatial buffering of glutamate and K^+^/Na^+^ ions, and the long-range redistribution and spreading of signaling molecules and Ca^2+^ waves [7,8]. Cx functionality also impacts myelin formation and maintenance [7]. Moreover, uncoupled connexins can work as plasma membrane hemichannels that are involved in the release to the extracellular milieu of small signaling molecules and gliotransmitters, such as ATP, glutamate, nicotinamide adenine dinucleotide (NAD), d-serine, and prostaglandins, in physiological and pathological conditions [9,10,11]. Functional alterations of the astrocyte gap junctions and Cx43 have been reported in several neuroinflammatory and neurodegenerative diseases and in brain tumors, indicating that these structures are crucial for the maintenance of the CNS homeostasis [5,11,12,13]. Recent studies unravel also significant expression and functional involvement of Cx43 at perivascular astrocytes forming the blood-brain barrier (BBB) [14,15].

The present study is aimed at elucidating the possible involvement of astrocytic Cx43/gap-junction dysfunctions in the pathogenesis of the rare megalencephalic leukoencephalopathy with subcortical cysts (MLC). Astrocytes are the cells primarily affected in MLC disease, showing defective regulation of cell volume changes and ion/water exchanges [16]. MLC patients develop brain edema, fluid cysts, myelin, and astrocyte vacuolation leading to severe brain damage, loss of motor functions, epilepsy, and cognitive decline [17,18,19]. To date, no therapy exists for MLC and therapeutic development is hindered by a lack of a thorough understanding of MLC molecular mechanisms. The disease is mainly caused by mutations in MLC1, an astrocyte-specific protein whose proper function is still unknown [20], and in GlialCAM, an adhesion-like protein facilitating MLC1 expression at the plasma membrane [21]. Both MLC1 and GlialCAM proteins are highly expressed at astrocyte end-feet contacting blood vessels and meninges and are particularly enriched at astrocyte-astrocyte junctions [22,23,24], where they interact with several pumps and ion channels ([25] and the references therein) and with the junctional proteins ZO-1 and Cx43 [24,26]. However, to date, nothing is known about the effects of MLC1 mutations on Cx43 and gap-junction functionality in astrocytes. We here investigated possible structural and functional relations between MLC1 and Cx43 by using an already characterized MLC cellular model represented by U251 astrocytoma cells overexpressing MLC1 wild type (WT) or carrying pathological mutations [27,28,29,30]. Our results revealed unexpected connections and regulations that provide new insights about MLC pathogenesis and the identification of possible therapeutic targets.

## 2. Materials and Methods

### 2.1. U251-MLC1 Cell Line Generation, Culture Conditions, and Treatments

For the generation of the U251 astrocytoma cell lines stably expressing MLC1 WT or carrying the c.178-10 t>a (Pt1) or c.177+1 g>t (Pt2) pathological mutations [31,32], retroviral vectors were constructed as described in Reference [27] using the MLC1 cDNAs derived from patient peripheral blood mononuclear cells (PBMC) that were available at the Bambino Gesù Hospital in Rome after obtaining patient fully informed consent. U251 cells were then infected with retroviral vectors expressing recombinant MLC1 cloned in frame with the X-press tag peptide (DLYDDDDK). The infected cell lines were then cultured and maintained in the selective medium Dulbecco’s modified Eagle’s high glucose medium (DMEM, Euroclone, Ltd., Milan, Italy) supplemented with 10% fetal bovine serum (FBS), (Gibco BRL, Paisley, UK), 1% penicillin/streptomycin (Sigma Ltd., Irvine, UK), and 600 mg/mL G418 (Gentamicin, Euroclone Ldt.) at 37 °C in a 5% CO_2_/95% air atmosphere, as previously described [27]. The c.178-10 t>a is a homozygous splice site mutation in intron 2, upstream of exon 3, that disrupts the sequence between intron 2 and exon 3, causing a complete skipping of exon 3 and the consequent aberrant translation of the MLC1 protein I and II transmembrane domains [31,32,33]. The c.177+1 g>t homozygous mutation hits the splice-donor GT sequence of exon 2, resulting in a TT transition. The Human Splicing Finder (HSF) bioinformatics tool predicts a donor splice site break in intron 2, leading to incorrect splicing of exon 2 and possible (total or partial) intron 2 retention. After the missing splice-donor sequence, intron 2 DNA sequence contains a premature in-frame stop codon that most likely leads to a truncated and rapidly degraded MLC1 gene product [31,32,33]. We also used U251 cells infected with an empty vector (Ø), already available in the laboratory, as the control cell line. For extracellular-signal-regulated kinase 1/2 (ERK1/2) inhibition experiments, cells were stimulated with 50 µM of PD98059 (PD) (Santa Cruz Biotechnology, Inc., Santa Cruz, CA, USA) for 1 hour (h) and used for immunofluorescence (IF) staining and Western blot (WB) analysis, as described below. Cycloheximide (CHX)/epidermal growth factor (EGF) treatment to assess Cx43 stability/degradation kinetics was carried out by stimulating cells with CHX (100 μg/mL, Sigma-Aldrich, Darmstadt, Germany) alone or in combination with EGF (200 ng/mL, Sigma Aldrich) for different time lengths (30 min, 1 h or 2 h). Similarly, treatment with EGF alone (200 ng/mL) was also performed on MLC1-expressing cell lines. After stimulation, cells were washed in phosphate buffered saline (PBS), collected by scraping, and centrifuged at 2700× *g* at 4 °C for 10 min. Cell pellets were solubilized and used for total protein extraction and WB analysis, as described below.

### 2.2. Total RNA Extraction and RT-PCR

Total RNA derived from U251 cells, both mock-infected (Ø) and expressing MLC1 WT or carrying the Pt1/Pt2 mutations, was purified using SV Total RNA Isolation System (Promega, Madison, WI, USA). One µg of total RNA was retrotranscribed, and PCR reactions for MLC1 and human β-actin were performed as previously described [23,29].

### 2.3. Immunofluorescence and Confocal Microscopy Analysis

For immunofluorescence staining, cells were grown subconfluent on polylysine-coated cover slips, fixed for 10 min with 4% paraformaldehyde (PFA), and washed with PBS. After 1 h of incubation with blocking solution (5% bovine serum albumin in PBS), cells were incubated overnight (ON) at 4 °C with the primary antibody (Ab) anti-connexin43 (Cx43) polyclonal (p)Ab (1:50, Abcam, Cambridge, MA, USA, recognizing the Cx43 C-ter) and were diluted in PBS and 0.025% Triton X-100 for 1 h at room temperature (RT) with the following primary Abs diluted in PBS and 0.025% Triton X-100: anti-Xpress monoclonal (m)Ab (1:50, ThermoFischer Scientific, Rockford, IL, USA), anti-EEA1 mAb (1:50, BD Transduction Laboratories, Lexington, KY, USA), anti-Rab11 mAb (clone47; 1:25, Millipore, Temecula, CA, USA), anti-Lamp-2 mAb (1:100, Abcam, Cambridge, MA, USA), and anti-GlialCAM pAb (1:50, Proteintech, Chicago, IL, USA). As secondary Abs, biotin-SP-AffiniPure goat anti-rabbit IgG H+L (4.3 μg/mL; Jackson Immunoresearch Laboratories, West Grove, PA, USA) followed by incubation with 2 μg/mL streptavidin-Tetramethylrhodamine (TRITC) (Jackson, USA) or Alexa Fluor 488 goat anti-mouse IgG (1:300, Invitrogen, Milan, Italy) were used. To stain actin filaments, a fluorescein (FITC)-conjugated phallacidin high-affinity F-actin probe (1:30, Invitrogen) was used. Coverslips were washed, sealed in Fluoroshield with 4′,6-diamidino-2-phenylindole (DAPI), (F6057, Sigma Aldrich), and analyzed with a laser scanning confocal microscope (LSM 5 Pascal, Carl Zeiss, Jena, Germany) or with a Leica DM2100 fluorescence microscope.

### 2.4. Protein Extract Preparation and Western Blotting

Cytosol and membrane (Triton-soluble) protein fraction from U251 astrocytoma cell lines were obtained as previously described [27,34]. For Triton-insoluble protein extraction, the insoluble pellets remaining after membrane protein extraction were left 15 min on ice in a solution containing 1% Triton X-100, 0.5% sodium deoxycholate, 150 mM NaCl, 10 mM (4-(2-hydroxyethyl)-1-piperazineethanesulfonic acid (HEPES) (pH 7.4), and protease inhibitor cocktail plus 1%SDS and then were sonicated for 10 min; maintained for 30 min on ice, as described in Reference [35]; and mixed with loading buffer. Protein samples were then subjected to SDS-PAGE using gradient (4–12%) pre-casted gels (Life Technologies, Grand Island, NY, USA), transferred to a nitrocellulose membrane, blocked 1 h with 7% dry milk, and blotted ON at 4 °C with the following primary Abs: anti-MLC1 pAb (1:1500, in-house generated), anti-Actin mAb (1:2000, Santa Cruz Biotechnology, Santa Cruz, CA, USA), anti-pERK1/2 (Thr202/Tyr204) pAb, (1:1000, Cell Signaling Technology, Danvers, MA, USA), anti-connexin43 (Cx43) pAb (1:3000, Abcam, Cambridge, MA, USA), anti-GlialCAM pAb (1:1000; Proteintech, Chicago, IL, USA), and anti-Xpress mAb (1:1000, ThermoFischer Scientific, MA, USA). After washings in tris buffered saline (TBS), membranes were incubated with horseradish peroxidase-conjugated anti-mouse or anti-rabbit Abs(1:5000; Biorad Laboratories, Hercules, CA, USA) for 1 h at RT. Immunoreactive bands were visualized using an enhanced chemiluminescence reagent (Pierce, ThermoFisher Scientific, Rockford, IL, USA), according to the manufacturer’s instructions, and exposed on a Bio-Rad ChemiDoc XRS system. Densitometric analyses of WB experiments were performed using NIH ImageJ software or Bio-Rad ChemiDoc XRS system. Quantification of protein loading content was carried out using a bicinchonic acid assay (BCA kit; Thermo Scientific, Waltham, MA, USA). Quantification of the WB bands was performed using the ImageJ software (National Institutes of Health, Bethesda, MD, USA).

### 2.5. Co-Immunoprecipitation Assay

Protein extract derived from U251 astrocytoma cell lines overexpressing WT MLC1 was obtained by solubilization of about 3 × 10^6^ cells in binding buffer [20 mM HEPES, pH 7.4, 150 mM NaCl, 1 mM Ethylenediaminetetraacetic acid (EDTA), 1 mM Ethylene glycol-bis(2-aminoethylether)-*N*,*N*,*N* (EGTA), 0.1% (*v/v*) Triton X-100, 0.1% (*v/v*) sodium deoxycolate, 0.1 mM phenylmethylsulfonyl fluoride (PMSF), and phosphatase inhibitor cocktail] for 1 h at 0 °C. After centrifugation at 10,000× *g* for 10 min, the supernatant was collected and subjected to protein content measurement using the BCA kit. Cell lysate (1.5 mg/mL) was incubated with 50% (*w/v*) protein A/G PLUS agarose beads (Santa Cruz Biotechnology, Santa Cruz, CA, USA) for 2 h at 4 °C, clarified by centrifugation, and incubated with the anti-Xpress mAb (1–2 µg/sample) ON at 4 °C. The immunocomplex was precipitated by the addition of 50% (*w/v*) Protein A/G PLUS agarose beads. The beads were extensively washed in binding buffer and TBS and processed for WB analysis as described above.

### 2.6. Measurement of Gap-Junction Permeability by Neurobiotin Microinjection Method

To evaluate the gap-junction permeability in U251 cells expressing WT and mutant MLC1, the whole-cell configuration of the patch-clamp technique was used to inject the gap-junction permeable Neurobiotin tracer (Vector Laboratories, Burlingame, CA, USA); 40,000 cells/well were seeded on poly-L-Lysine coated glass coverslips in 12-well plates and cultured as above. After 4 days, coverslips hosting cells were mounted in the recording chamber and superfused with the following solution (mM): NaCl 140, KCl 5, CaCl_2_ 2.5, MgCl_2_ 1, HEPES 10, and glucose 10; pH was adjusted to 7.4 with NaOH. The patch-clamp setup included an Axopatch 200B amplifier (Molecular Devices, San Jose, CA, USA) supported by Clampex 11 software (Molecular Devices) and a phase contrast microscope (Leica, DMLA, Wetzlar, Germany). Patch pipettes were obtained by a Narishige vertical puller from thin-wall borosilicate glass capillaries (World Precision Instruments). Pipettes were then filled with an internal solution of the following composition (mM): K-Gluconate 130, CaCl_2_ 1, EGTA 11, HEPES 10, and Mg-ATP; pH was adjusted to 7.2 by adding KOH. Neurobiotin 0.2% (*w/v*) was added in the internal solution. Electrodes with a resistance of 3–4 MOhm were used. Once the whole-cell configuration was achieved, access resistance was monitored along with the experiment for verifying the efficiency of injection. The whole-cell configuration was maintained for 10 min in voltage-clamp mode at a zero current holding potential (around −30 mV); then, the electrode was gently removed. To minimize injection variability, the above procedure was rigorously applied. After the injection procedure, coverslips were fixed in 4% PFA for 20 min at RT and incubated in 2 μg/mL streptavidin-TRITC (Jackson, USA) and 0.025% Triton-X-100 in PBS for 45 min. Finally, coverslips were washed, sealed in Fluoroshield with DAPI (F6057 Sigma Aldrich), and analyzed with a Leica DM4000B fluorescence microscope equipped with a DFC420C digital camera and Leica Application Suite Software (260RI) (Leica, Wetzlar, Germany). Imaging analysis was performed using the freeware ImageJ (NIH, https://imagej.nih.gov/ij/index.html). Regions of interest (ROIs) of all the cells in the fields were obtained according to DAPI channel and to the bright field image of the same field for each condition; then, the center of mass of each ROI was calculated and stored as a mask. Tumor-derived cells such as U251 often show multiple nuclei; the identification of one nucleus per cell was made possible by superimposing DAPI fluorescence to the corresponding bright field image. The resulting mask was then applied to the Neurobiotin images for these purposes: i) counting the number of diffused cells, ii) calculating the distance of the Neurobiotin loaded cells from the injection site, and iii) calculating the fluorescence intensity of Neurobiotin diffused cells. Euclidean distances from the injected cell of Neurobiotin positive cells were measured according to each center of mass, for each condition. The number of neurobiotin-positive cells were pooled and distributed as a boxplot. ANOVA hypothesis testing was used to test statistical significance.

### 2.7. Ethidium Bromide (EtBr) Uptake Assay

To monitor hemichannel function in U251 astrocytoma cell lines expressing WT and mutated MLC1, EtBr uptake assay was performed as described in Reference [36]. Briefly, 50,000 cells were seeded in 12-well plates and cultured as described above. After 24 h, cells were washed and exposed to 15 μM EtBr (Abcam) for 10 min at 37 °C. EtBr is impermeable through membrane but can transit through hemichannels and becomes more fluorescent after binding to DNA. After 10 min exposure to EtBr, astrocytes were washed in PBS and fixed in 4% PFA. Coverslips were washed and sealed in Fluoroshield with DAPI (F6057, Sigma Aldrich). Images were taken by a LSM980 Zeiss confocal microscope (Zeiss, Germany) using a (Zeiss) planar objective at 20×. Excitation light was obtained by a Laser Dapi 408 nm for DAPI and a Diode Laser HeNe (543 nm) for EtBr. DAPI emission was recorded from 415 to 485 nm and from 543 to 694 nm for EtBr. For each experimental point, 5 microscopic fields were captured discretionally. Images were analyzed by Zen software (Blu version, Zeiss, Germany) for image analysis. EtBr mean fluorescence intensity was evaluated in single cells. Three experiments were carried out on in triplicate.

### 2.8. Statistical Analysis

All the statistical analyses were performed using GraphPad prism software. Results were expressed as mean values ± standard error of the mean (SEM). Data were first subjected to normality test (D’Agostino and Pearson Omnibus Normality test). When data followed a normal distribution, a Student-t test was applied; otherwise, nonparametric tests, such as Kruskal–Wallis test, were used, followed by Dunn’s Multiple Comparison test post hoc test when necessary. The test used is indicated in the corresponding legend of each figure. *p* values are * *p* < 0.05, ** *p* < 0.01 and *** *p* < 0.001. Colocalization analysis of immunofluorescence images was performed with ImageJ software using the colocalization plugin Coloc2 (https://imagej.net/Coloc_2). Colocalization values (Pearson and Manders’ coefficients) were calculated from 5 representative images derived from three independent experiments.

## 3. Results

### 3.1. Analysis of Cx43 Expression and Localization in U251 Astrocytoma Cells Overexpressing MLC1 Wild Type or Carrying Pathological Mutations

To disclose the structural and functional relationships between MLC1 and Cx43 in astrocytes and the possible effects of MLC1 pathological mutations on them, we first generated U251 astrocytoma cells stably producing recombinant MLC1 wild type (WT) or MLC1 carrying two mutations found in MLC patients: c.178-10 t>a (Pt1) and c.177+1 g>t (Pt2) [31,32]. To this aim, retroviral vectors expressing WT or mutated MLC1 were constructed as described in the Material and Methods section and used to stably infect the U251 mother cell line. We then characterized MLC1 mRNA and protein expression/distribution in the newly generated cell lines. *MLC1* gene sequence analysis and bioinformatics prediction indicate that both Pt1 and Pt2 mutations affect donor/acceptor splicing sites [31,32,33] potentially causing (i) the skipping of exon 3 and the expression of shorter MLC1 mRNA and protein in cells carrying the Pt1 mutations, and (ii) the retention (partial or total) of the intron 2 and the transcription of a longer MLC1 mRNA in Pt2 cells when compared to WT MLC1 expressing cells. The consequent appearance of a premature in-frame stop codon, localized in the proximity of the Pt2 mutation site, is thought to generate an early truncated and quickly degraded MLC1 protein product in cells expressing this latter mutation (see the Materials and Methods section for further details). WB analysis of MLC1^+^ cell lines showed that the Pt1 mutation led to (i) a decrease of the monomeric form of the MLC1 protein [the 30–36 kDa molecular weight (MW) form], either in the cytosol and in the membrane protein fractions, and the lack of the dimeric, membrane associated form (the 60–64 kDa MW form) that had been previously described in astrocytes and brain tissue [23,27,37] and (ii) a decrease of the MW of the mutant protein, as predicted by bioinformatics tools (Figure 1A). When the same analysis was applied to Pt2 mutant cells, no MLC1-specific bands were detected in the cytosolic and membrane cell extracts, making likely the assumption that this mutation causes the expression of an early truncated and rapidly degraded gene product (Figure 1A). No MLC1 protein expression was observed in U251 cells infected with an empty vector (Ø, Figure 1A). RT-PCR analysis was then performed to monitor MLC1 mRNA expression in the different U251 cell lines. As expected, very low levels of MLC1 mRNA were observed in the mock-infected cells (Ø), representing the endogenous MLC1 mRNA expressed in the U251 cells, as previously reported [23]. Much higher levels of MLC1 mRNA were detected in cells expressing WT MLC1 and the Pt1 mutant without quantitative differences between them and in accordance with the high efficiency of the cytomegalovirus (CMV) promoter controlling recombinant MLC1 expression in the MLC1^+^ cell lines. Conversely, low levels of MLC1 mRNA were found in Pt2 cells, probably due to RNA messenger instability caused by the splicing site mutation generating an aberrant transcript. Differences of mRNA size among WT and mutant MLC1-expressing cells further reinforced the assumption that genetic defects lead to the skipping of exon 3 and intron 2 retention in Pt1 and Pt2, respectively. IF staining supported WB and RT-PCR results showing that, contrary to the plasma membrane and endoplasmic reticulum (ER) distribution observed in WT cells (Figure 1A and Appendix A), MLC1 carrying the Pt1 mutation was barely detectable only in the cytoplasmic areas. No MLC1 signal was detected in the Pt2 mutant expressing cells (Figure 1C) and mock-infected cells (Appendix A).

We then monitored Cx43 expression and distribution in the MLC1-expressing cell lines. To this aim, cytosolic and membrane Triton-soluble proteins were extracted and analyzed by WB. We also investigated Cx43 distribution in the Triton-insoluble fractions (obtained by sonication of membrane protein extracts in the presence of SDS), where gap-junction proteins are partitioned. These experiments revealed that, in MLC1 mutant expressing cells, Cx43 was preferentially localized in the cytosolic fractions when compared to WT cells (Figure 2A), while no significant differences were observed in the membrane protein extracts among the three cell lines (Figure 2B). When the Triton-insoluble, gap-junction-enriched fractions were analyzed, higher levels of Cx43 were found in WT MLC1^+^ cells when compared to cells expressing Pt1 and Pt2 mutants (Figure 2C).

In accordance to WB results, IF staining showed the typical distribution of Cx43 in discrete areas of the plasma membrane where cell-cell contacts (junctions) occur in WT MLC1-expressing cells (Figure 3A, arrows). Conversely, in mutant cells, Cx43 signals were mainly detected in large aggregates in the cytoplasmic areas and at reduced levels at the cell–cell junctions (Figure 3B,C, arrowheads) when compared to WT MLC1-expressing cells. Noteworthily, colocalization of MLC1 and Cx43 was also observed in some intercellular contacts in WT MLC1^+^ cells (Figure 3D,E arrows), as confirmed by the quantification of colocalized fluorescent signals (Figure 3F). A strong distribution of Cx43 protein in perinuclear regions was also observed in all the cell lines analyzed, as already reported in other glioma cells [38]. A greater Cx43 distribution in the cytoplasmic areas than in the plasma membranes is generally observed in cultured cells where gap-junction formation occurs to a lesser extent than in tissues.

### 3.2. MLC1 Expression Alters Cx43 Intracellular Trafficking in U251 Cells

Both WB and IF analysis of Cx43 distribution in MLC1-expressing astrocytoma lines indicated that WT MLC1 expression favored the partitioning of Cx43 in the Triton-insoluble gap-junction fraction. By contrast, in cells lacking functional MLC1, higher levels of Cx43 were observed in the cytoplasmic compartment. These results are suggestive of a possible effect of WT MLC1 expression on Cx43 intracellular trafficking and stabilization at plasma membrane gap-junctional areas. To test this hypothesis and to assess possible differences of Cx43 trafficking in the MLC1-expressing cell lines, we performed co-immunostaining of Cx43 with EEA1, Rab11, or Lamp2 Antibodies (Abs) that recognize early and recycling endosomes and lysosomes, respectively. These experiments showed higher levels of Cx43/EEA1/Rab11 colocalization in Pt1 and Pt2 cells when compared to WT MLC1-expressing cells (Figure 4A–C), suggesting that Cx43 undergoes a decrease of internalization rate when WT but not mutated MLC1 is expressed. On the contrary, no significant differences were observed when Cx43/Lamp2 colocalization was evaluated (Appendix A).

To further substantiate this observation, we treated U251 cell lines with EGF (200 ng/mL), a known stimulator of Cx43 endocytosis [39], in the presence of 100 µg/mL of cycloheximide (CHX), a protein synthesis inhibitor already used to study Cx43 protein stability [40], for 30 min, 1 h, and 2 h and assessed Cx43 turnover and degradation kinetics by WB analysis. These experiments showed that Cx43 had an increased stability in WT MLC1^+^ cells when compared to mutant cell lines, as shown by its slower rate of degradation, particularly at 30 min and 1 h, upon inhibition of translational elongation by CHX (Figure 5A,B). No significant differences of Cx43 expression/degradation were observed when the U251 cell lines were treated with CHX or EGF alone (Appendix A).

### 3.3. MLC1 Expression Affects ERK-Mediated Cx43 Phosphorylation in U251 Cells

The results described above revealed that WT MLC1 expression slows down Cx43 turnover by the endosomal pathway, thus favoring the Cx43 stabilization in gap-junction compartments. To verify whether the MLC1-mediated regulation of Cx43 trafficking was due to a direct interaction between the two proteins, as observed previously for the GlialCAM/Cx43 complex [40], we performed an immunoprecipitation (IP) assay on WT MLC1-expressing cells using the anti-Xpress mAb. As shown in Appendix A, the anti-Xpress mAb was effective at immunoprecipitating both MLC1 monomeric and dimeric forms and the MLC1 direct interactor GlialCAM in WT and, to a lower extend, in Pt1 cells. Conversely, no immunoprecipitated proteins were found in samples derived from Pt2 cells. These latter cells were then used as a negative control. In all the cell lines analyzed, Cx43 was not found in the co-immunoprecipated proteins (Appendix A), thus ruling out the possibility that MLC1 stabilization of Cx43 in astrocytoma cells was due to a direct protein–protein interaction. To further elucidate the molecular mechanisms leading to MLC1-mediated Cx43 stabilization, we speculated that MLC1 could affect Cx43 trafficking by influencing Cx43 phosphorylation. Indeed, Cx43 is a highly phosphorylated protein and substrate of different types of kinases (protein kinase C (PKC), protein kinase A (PKA), casein kinase (CK1), mitogen-activated protein kinase (MAPK), and Src) which are able to regulate its compartmentalization and functionality [41,42]. In particular, the increased phosphorylation by MAPK/ERK1/2 kinases was found to decrease Cx43 gap-junction formation via activation of protein endocytosis [42]. Since in U251 cells we recently revealed MLC1-mediated inhibition of phospho-ERK1/2 (pERK1/2) [29,30] that, in U251 cells, is constitutively activated [43], we hypothesized that MLC1 might indirectly alter Cx43 trafficking by inhibiting pERK1/2 activation in astrocytoma cells. To test this hypothesis, we first monitored pERK1/2 expression in all the MLC1-expressing cell lines. WB analysis of the cytoplasmic fractions showed the inhibition of pERK1/2 in WT MLC1 cells when compared to the Pt1 and Pt2 cells (Figure 6), as previously described [29].

Thus, to figure out whether differences of Cx43 trafficking and partitioning in gap-junction areas were actually due to the inhibition of pERK1/2 in WT MLC1-expressing cells, we treated the U251 cell lines with PD98059 (PD), a specific inhibitor of ERK1/2 kinases. We analyzed the consequences of this treatment on pERK1/2 activation and Cx43 distribution in gap-junction compartments by WB. These experiments confirmed that 1 h of PD stimulation at a concentration of 50 µM was able to inhibit pERK1/2 activation in the cytosolic fraction of all the U251 cell line analyzed (Figure 7A) and revealed that pERK1/2 inhibition caused an increase of Cx43 distribution in the Triton-insoluble fractions of Pt1 and Pt2 cells when compared to unstimulated cells (Figure 7B). No significant alterations of Cx43 expression were observed in protein extracts derived from WT MLC1 after PD stimulation (Figure 7B).

The increased distribution of Cx43 in the Triton-insoluble compartments of mutant cells in response to pERK1/2 inhibitor treatment supports the assumption that MLC1 protein expression favors Cx43 distribution in gap junctions by inhibiting ERK1/2-mediated Cx43 phosphorylation.

### 3.4. Effects of MLC1 Expression on CX43 Functions

To explore whether the MLC1 regulation of Cx43 trafficking impacted Cx43 function, we performed two different assays to monitor Cx43 gap junction and hemichannel function in the MLC1-expressing cell lines: the neurobiotin transfer assay and the ethidium bromide (EtBr) uptake assay.

*i)* Neurobiotin transfer assay: given its low molecular weight and charge, neurobiotin freely moves through gap-junction channels, but it is completely unable to cross membrane lipid bilayers. Thanks to these features, neurobiotin is widely used for studying cell morphology and for evaluating intercellular communication through gap-junction coupling in different type of cells, including U251 astrocytoma cells [44]. Taking advantage of the imaging analysis capabilities of ImageJ software, we analyzed neurobiotin and DAPI fluorescence images. We calculated cell mass centers and boundaries, distances from single cells, and average fluorescence intensities of single cells with the goal of drawing the most descriptive and informative picture of neurobiotin diffusion in the different cell lines. Among the parameters we evaluated, distance from injection and fluorescence intensity did not help to depict differences among the cell lines. In our view, this was due to variability in cell size, both within clones and among the different clones, and of the inverse but not constant relationship between higher fluorescence intensity and number of positive interconnected cells, therefore biasing the interpretation of the results. The number of neurobiotin positive cells was the parameter most clearly showing the difference among the MLC1 expressing cell lines. In keeping with biochemical Cx43 protein quantification by WB and with subcellular distribution analysis by IF experiments, the WT MLC1 cell line showed a significantly higher number of neurobiotin-positive cells around the injected ones when compared to both MLC1 mutant-expressing cells Pt1 and Pt2 (Figure 8 A,B). Worth noting, among mutant cells, Pt2 showed the lowest number of neurobiotin-positive cells, in strict correlation with biochemical/IF data, showing the lowest levels of Cx43 expression in Triton-insoluble fractions and plasma membrane gap-junction areas in this cell line. Boxplot of neurobiotin-positive cells also revealed a degree of variability of positive cells in the different images (injections) that may be due to variability in the quantitative expression of MLC1 in the single cell injected. However, the statistical analysis of the total set of neurobiotin-injection experiments clearly showed that intercellular communication through gap junctions was more efficient in WT compared to mutant MLC1-expressing cells.

*ii)* Ethidium bromide uptake assay: since Cx43 can form unopposed hemichannels exerting different functions in pathophysiological conditions [11], we also monitored Cx43 hemichannel function by using EtBr uptake assay as previously described [36]. After 24 h from seeding, cells were incubated for 10 min with EtBr and then fluorescence emission was recorded and quantified by a confocal microscope, as described in the Material and Methods section. This analysis did not reveal significant differences among the WT MLC1 and Pt1 and Pt2 mutant-expressing cells (Appendix A), indicating that the Cx43 hemichannel function is not influenced by MLC1 expression.

## 4. Discussion

In the present work, we analyzed for the first time the relationships between the gap-junction protein Cx43 and MLC1, the astrocytic protein whose mutations are responsible for megalencephalic leukoencephalopathy with subcortical cysts (MLC). This disease is a rare congenital leukodystrophy characterized by early onset of macrocephaly, progressive motor dysfunction, and mild cognitive deterioration. Affected children develop cerebellar ataxia and spasticity, and most of them become wheelchair-dependent during adolescence. Seizures and aggravation of clinical conditions after minor head trauma or fever are also observed in MLC patients [45]. No treatment is currently available for this disease, and patients undergo only supportive therapies. The development of possible therapeutic interventions is hampered by the lack of knowledge about MLC molecular pathogenesis and the proper function of MLC1, the protein, found mutated in 80% of patients. The discovery of a second protein linked to MLC, the MLC1 molecular interactor GlialCAM [21], allowed identifying two disease variants, MLC2A and B, caused by recessive and dominant mutations in GlialCAM that show some differences in the disease progression [19,46,47]. With the aim of shedding light on MLC molecular pathogenesis, we here investigated Cx43 properties in U251 astrocytoma cell lines overexpressing MLC1 WT or carrying pathological mutations. Several experimental evidences from in vivo and in vitro studies gave hints on possible structural and functional connections between MLC1 and Cx43. In human and mouse brains, the MLC1/GlialCAM protein complex is highly expressed at perivascular astrocyte end-feet contacting the BBB and is particularly enriched at astrocyte–astrocyte junctions, where it establishes interactions with ZO-1 and Cx43 [24,40]. The diffuse brain edema characterizing the magnetic resonance imaging (MRI) of MLC patients and astrocyte vacuolation/swelling observed by histological analysis of brain tissue (see Reference [46] and the reference therein) along with alterations of chloride, potassium, and calcium fluxes reported in MLC disease models [27,48,49,50] suggest that intercellular communication and glial syncytium properties might be affected directly or indirectly by MLC1 mutations. A further support to this hypothesis is provided by the finding that in astrocytoma cells GlialCAM pathological mutations disrupt GlialCAM/Cx43 interactions, decreasing Cx43 stability at the plasma membrane [40]. By using newly generated U251 astrocytoma cell lines expressing WT or mutant MLC1, we found that WT but not mutated MLC1 expression favors Cx43 partitioning in the Triton-insoluble membrane protein compartment that corresponds to connexin-assembled gap-junctional plaques. Conversely, Cx43 levels are higher in the cytoplasmic organelles of cells expressing the MLC1 mutants Pt1 and Pt2 when compared to WT cells, clearly indicating a possible role for MLC1 in the control of Cx43 trafficking. Consistent with this, IF experiments revealed the presence of some Cx43^+^ large circular aggregates localized in sub-plasmalemmal areas in MLC1 mutant-expressing cells, particularly Pt1 cells, that may correspond to annular junction structures formed during gap-junction internalization (Figure 4) [51]. CHX experiments confirmed this speculation, demonstrating that MLC1 expression in U251 astrocytoma cells is able to increase stability of Cx43 proteins at the plasma membrane, as indicated by its slower degradation time in WT when compared to MLC1 mutant-expressing cells. In an attempt to disclose the molecular mechanism underlining the MLC1-mediated effect on Cx43 distribution in astrocytes, we first evaluated if a direct protein–protein interaction mechanism occurred, as observed for Cx43/GlialCAM. However, MLC1/Cx43 co-immunoprecipitation experiments failed, bringing into play other indirect molecular events. We then verified the possibility that MLC1 influenced Cx43 distribution and functionality by affecting its phosphorylation. Indeed, the majority of connexins are highly phosphorylated proteins and many protein kinases have been found to phosphorylate Cx43 on multiple serine residues [42], including PKC, ERK1/2, CK1, and src kinase in different types of cells (see References [41,42] and the references therein). Changes in connexin phosphorylation status are closely associated with changes in gap-junction assembly, stability/degradation, and channel properties. Although Cx assembly and turnover are very complex processes involving many phosphorylation/dephosphorylation steps that are still incompletely known, several experimental evidences indicated that Cx43 phosphorylation by ERK1/2 and PKC on S262 and S279/282 amino acid residues induces channel closure and internalization, leading to a rapid shutdown of intercellular communication (see References [41,42] and the references therein). Considering the MLC1-mediated inhibition of ERK1/2 activation that we recently demonstrated in astrocytes [29,30,52], we thought possible that MLC1 hampered ERK1/2-induced phosphorylation of Cx43 and its consequent internalization in WT MLC1-expressing cells. To verify this possibility, we evaluated the effects of a specific pERK1/2 inhibitor (the PD98059) on Cx43 partitioning in Triton-insoluble compartments, finding out that pERK1/2 inhibition rescued Cx43 distribution in gap junctions in cells expressing the MLC1 mutants. These results allowed identifying Cx43 as a target protein of the MLC1-mediated inhibition of pERK1/2 in astrocytes, similar to what was previously observed for the LRRC8C subunit of the volume-regulated anion channel (VRAC) [52]. Most importantly, the analysis of gap-junction functionality in the MLC1-expressing cells performed by the use of the permissive neurobiotin tracer [44] indicated that MLC1 expression favors junctional communication and greater neurobiotin intercellular spread in WT MLC1 cells when compared to mutant-expressing cells after single cell injections. These findings highlight the functional relevance of the MLC1-mediated Cx43 regulation described here. By combining our biochemical and functional results with the known GlialCAM/Cx43 molecular relationships [40], we hypothesized that MLC/GlialCAM complex exerts a double-step regulation on Cx43 trafficking and gap-junction functionality. After the GlialCAM-induced translocation and stabilization of Cx43 at the plasma membrane, MLC1 would hamper Cx43 internalization and channel closure by inhibiting/lowering Cx43 phosphorylation by ERK1/2 kinase, thus sustaining gap-junction communication among astrocytes. A schematic representation of this supposed molecular mechanism is shown in Figure 9. In our cell system, we cannot evaluate the contribute of endogenous GlialCAM, since in U251 cells, GlialCAM expression levels are very low (Appendix A), as already described [53], and probably they are not sufficient to influence Cx43 trafficking and stabilization. Moreover, no differences in GlialCAM expression have been observed after PD treatment (Appendix A). Our data also revealed that MLC1 expression does not alter Cx43 hemichannel function measured by the EtBr uptake assay, indicating that the MLC1-mediated effect on Cx43 is specifically restricted to Cx43 function in gap-junctional coupling. From the pathological point of view, the present results add a new piece of information to the complex scenario of MLC molecular pathogenesis. Gap-junction-coupled astrocyte end-feet form an extended perivascular syncytium fully covering the vascular surface [15]. Due to the highly enriched distribution of MLC1/GlialCAM complex and also Cx43 at this compartment [21,23,54], it is conceivable that alterations of Cx43 turnover/functionality due to phosphorylation changes can affect astrocyte permeability and communication properties, leading to deleterious consequences on BBB functions. Indeed, increased ERK1/2-mediated phosphorylation of Cx43 has been observed in brain pathological conditions and knocking down ERK1/2 by siRNA or by inhibition of its activity protected BBB integrity and prevented brain damage [55]. Altered astrocyte connectivity could also explain astrocyte end-feet swelling and the diffuse cerebral edema observed in MLC patients. A recent study indicating that loss of function of MLC1 leads to dysregulation of potassium ion (K^+^) homeostasis [49] suggests that, after astrocyte-mediated K^+^ uptake occurring at synaptic levels, defects in Cx43/gap junction coupling may affect K^+^ spatial buffering and expulsion at BBB sites [56]. Worth noting, Cx43 is needed for astroglial perivascular connectivity during postnatal BBB maturation [15,57], a process where also the MLC1/GlialCAM complex is involved [58]. These observations prompted us to hypothesize that alterations of MLC1/GlialCAM/Cx43 relationships during postnatal BBB maturation phases have a role in the development of edema and macrocephaly occurring in MLC patients after birth. Although our knowledge on MLC molecular mechanisms is growing, some questions on MLC1 proper function and on the spatial and temporal regulations/interrelations among the processes described above are still unresolved, hampering the identification of possible therapeutic strategies for this disease. In the present work, we shed lights on ERK1/2 kinases as new key elements in MLC molecular pathogenesis. Extracellular-signal-regulated kinases (ERK) are important components of the Ras-Raf-MEK-ERK signaling pathway that mediates intracellular stimuli transduction and gene expression. ERK1/2 kinases take part in the regulation of a variety of cell activities, such as cell proliferation, migration, and differentiation, in different types of cells, including astrocytes, particularly during development and in response to damaging signals, like inflammation or oxidative stress [59]. Our previous data indicating that MLC1 inhibits ERK1/2 activation in human astrocytoma cells, in primary mouse astrocytes [29,30,52], and, more recently, in astrocytes differentiated from patient-inducible pluripotent stem cells (iPSC, manuscript in preparation) strongly support the significant role played by these kinases in MLC brain dysfunctions. Further investigations are needed to explore the effect of ERK1/2 inhibition in MLC disease models. Overall, data presented here revealed Cx43 as a new player in MLC pathogenesis and MLC1 as a new regulator of Cx43 phosphorylation and suggested a target molecule to be explored for therapeutic purposes. In light of recent findings indicating that in some MLC patients the pathological process is partially reversible [19], the identification of a potential therapeutic target may strongly help the development of pharmacological strategies to arrest disease progression and ameliorate patient quality of life.

## Figures and Tables

**Figure 1 cells-09-01425-f001:**
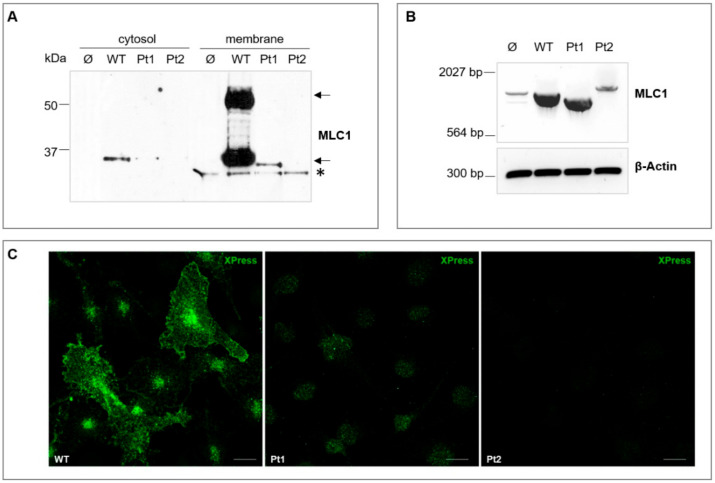
Analysis of MLC1 expression and localization in U251 astrocytoma cell lines infected with empty vector (Ø) or with vectors expressing wild type (WT) or mutant MLC1 (Pt1 and Pt2): (**A**) Western Blot (WB) analysis of protein extracts (40 µg) from U251 cell lines using anti-MLC1 pAb shows the presence of both the dimeric membrane form (~60kDa molecular weight, MW) and monomeric cytosolic form (~30kDa MW) of WT MLC1 (arrows). Pt1 mutation (c.178-10 t>a) leads to a strong reduction of MLC1 expression in the cytosolic/membrane fractions and inhibits MLC1 dimer formation. A decrease of the Pt1 mutant protein MW is also observed. No MLC1-specific bands are detected in both cytosolic and membrane protein fractions derived from cells expressing the Pt2 mutation (c.177+1 g>t). The asterisk marks an unspecific protein recognized by the anti-MLC1 pAb in all the membrane extracts. MW markers are indicated on the left (kDa). (**B**) RT-PCR analysis to monitor MLC1 mRNA expression in the U251 cell lines shows low levels of the endogenous MLC1 mRNA in mock-infected U251 cells (Ø). Much higher levels of MLC1 mRNA are observed in cells expressing WT and Pt1 MLC1. Low amounts of MLC1 mRNA are found in Pt2 cells. Differences of MLC1 mRNA size among WT, Pt1, and Pt2 cells are due to MLC1 genetic mutations causing splicing site defects that lead to the skipping of exon 3 (Pt1) and intron 2 retention (Pt2), respectively. Beta-Actin represents the housekeeping gene used to normalize mRNA levels among samples. (**C**) Immunofluorescence (IF) staining of U251 cell lines to analyze MLC1 localization was performed with anti-Xpress mAb (green) recognizing the Xpress epitope cloned in frame with the MLC1 sequence. MLC1 localizes at the plasma membrane and perinuclear areas in MLC1 WT cells. In Pt1 cells, mutated MLC1 fails to reach the plasma membrane and is retained in the cytoplasmic perinuclear areas. No signal is observed in Pt2 mutant-expressing cells, according to WB results. Scale bars = 20 µm.

**Figure 2 cells-09-01425-f002:**
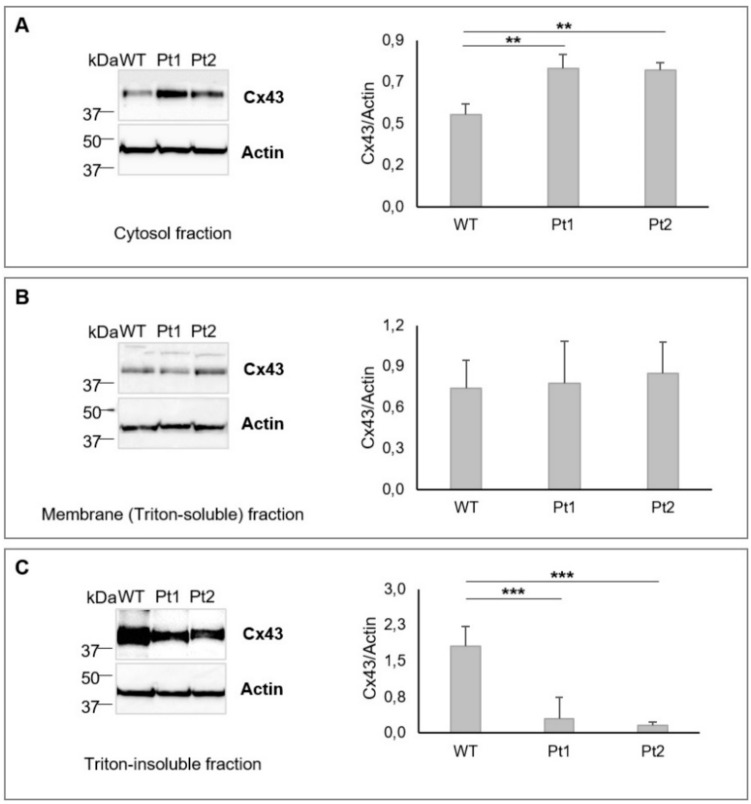
Cx43 protein expression in cytosolic and membrane Triton-soluble and Triton-insoluble fractions of U251 astrocytoma lines overexpressing MLC1: (**A**,**B**) WB analysis performed on 40 µg of protein extracts shows that Cx43 is significantly more expressed in the cytosolic compartments of Pt1 and Pt2 mutant cells when compared to WT MLC1 expressing cells (**A**) while no significative differences are observed in the membrane protein (Triton-soluble) extracts (**B**). (**C**) The analysis of Triton-insoluble fraction (60 µg of protein extracts) reveals significantly higher Cx43 protein levels in WT MLC1^+^ cells when compared to cells expressing Pt1 and Pt2 mutations. Actin is used as a loading control. Molecular weight markers are indicated on the left (kDa). One representative experiment out of three performed is shown. Densitometric analysis of the Cx43 protein band normalized with the amount of actin is evaluated for each cell fraction. The bar graphs represent means ± SEM of three experiments. Statistical difference was calculated using the Kruskal–Wallis test (** *p* < 0.01 and *** *p* < 0.001).

**Figure 3 cells-09-01425-f003:**
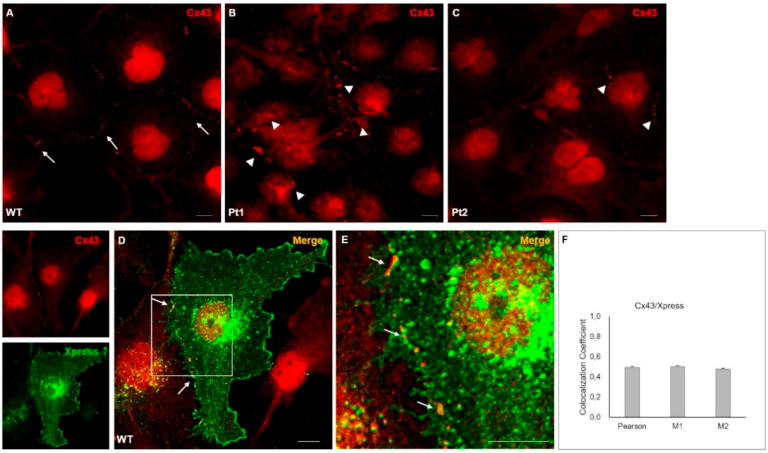
Cx43 protein localization in U251 astrocytoma cells overexpressing MLC1 WT or carrying the pathological mutations Pt1 and Pt2: (**A**) IF staining of WT MLC1-expressing cells by using the anti-Cx43 pAb (red) shows Cx43 protein distribution in discrete areas of the plasma membrane where cell-cell contacts occur (arrows). (**B**,**C**) In Pt1 and Pt2 mutant cells, Cx43 signal (red) is mainly detected in large aggregates, sometimes with a rounded shape, in the cytoplasm and scarcely at intercellular junctions (arrowheads). In all the cell lines analyzed, Cx43 is strongly enriched in perinuclear regions. (**D**) Double immunofluorescence staining of WT MLC1-expressing cells with anti-Cx43 pAb (red) and anti-Xpress mAb (green) recognizing MLC1 shows colocalization of MLC1 and Cx43 in some cell-cell contact areas (arrows). (**E**) High magnification of the inset in Figure 3D shows Cx43 (red)/MLC1 (Xpress, green) colocalization areas (arrows). Scale bars = 20 µm. (**F**) Cx43/MLC1 colocalization (according to the image in Figure 3D) was calculated as described in the Materials and Methods section. Pearson correlation coefficient and Manders’ correlation coefficients M1 and M2 are shown. Values are given as mean ± SEM of 5 representative images from each of the three independent experiments performed.

**Figure 4 cells-09-01425-f004:**
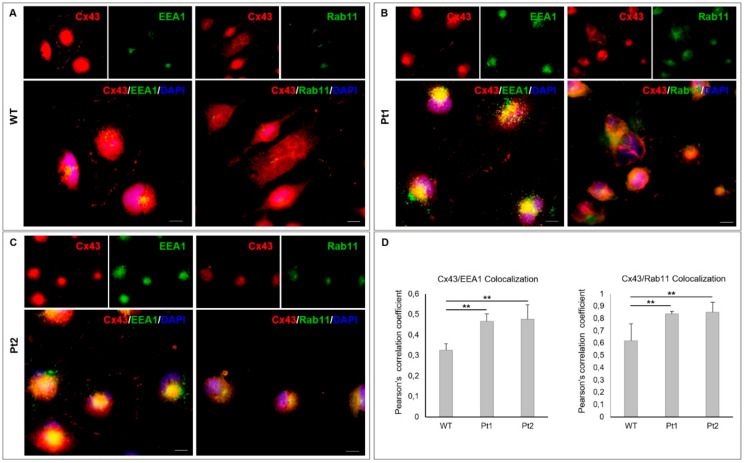
Analysis of Cx43 intracellular trafficking in U251 astrocytoma cells overexpressing MLC1 WT or carrying Pt1 and Pt2 pathological mutations: (**A**–**C**) Double immunofluorescence staining with anti-Cx43 pAb (red) and anti-early endosome antigen (EEA1) mAb (green) or anti-Rab11 mAb (green), which recognize early and recycling endosomes, respectively, indicates lower levels of Cx43/EEA1/Rab11 colocalization (yellow) in WT MLC1 cells (**A**) when compared to Pt1 and Pt2 expressing cells (**B**,**C**). Scale bars = 20 µm. (**D**) Colocalization analysis was performed using ImageJ and the Coloc2 plugin (https://imagej.net/Coloc_2). Cx43 colocalization with EEA1 or Rab11 was determined by the Pearson’s correlation coefficient. The average Pearson’s correlation coefficients ± SEM were calculated from five replicate experiments. Statistical analysis was performed using ANOVA followed by the Kruskal-Wallis test (** *p* < 0.01).

**Figure 5 cells-09-01425-f005:**
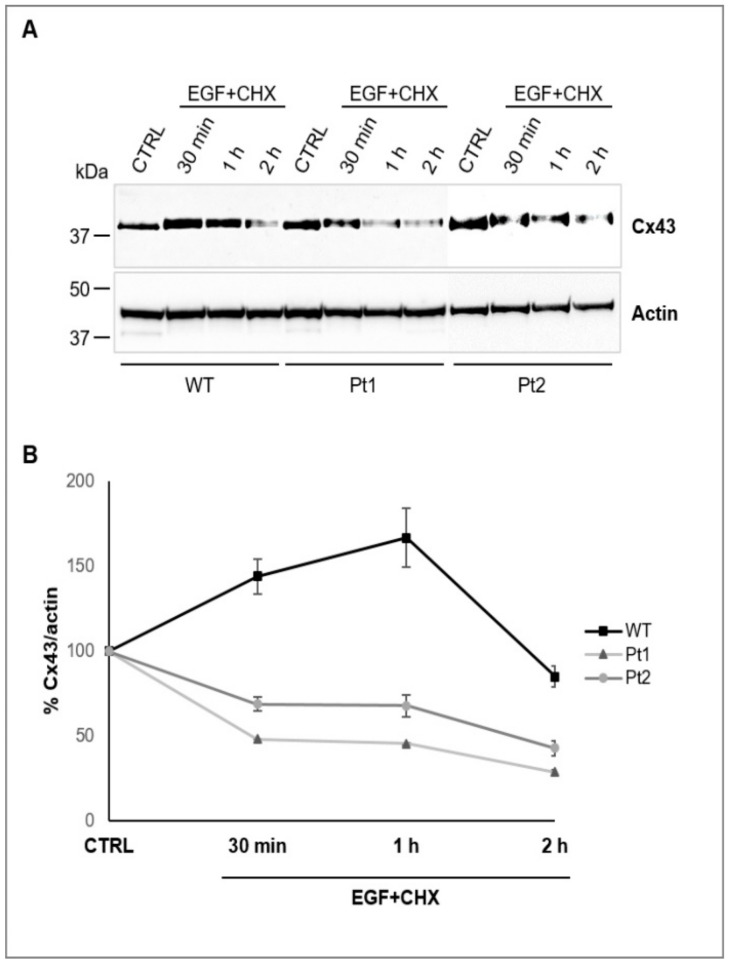
WT MLC1 increases Cx43 protein stability. (**A**) WB analysis of protein extracts (40 µg) of U251 astrocytoma cells overexpressing MLC1 WT or carrying Pt1 and Pt2 pathological mutations, untreated (CTRL) or treated with EGF (200 ng/mL) in the presence of 100 µg/mL of cycloheximide (CHX) for 30 min, 1 h and 2 h reveals differences in Cx43 degradation kinetics among the cell lines analyzed. Cx43 protein levels decrease slower in WT MLC1-expressing cells when compared to the Pt1 and Pt2 mutant cell lines particularly after 30 min and 1 h of EGF/CHX treatment. Actin is used as a loading control. Molecular weight markers are indicated on the left (kDa). One representative experiment out of three performed is shown. (**B**) Densitometric analysis of Cx43 protein bands normalized with actin protein levels in the same samples: Data are expressed as a percentage of the value measured in control untreated cells (100%). Means ± SEM of three experiments are shown.

**Figure 6 cells-09-01425-f006:**
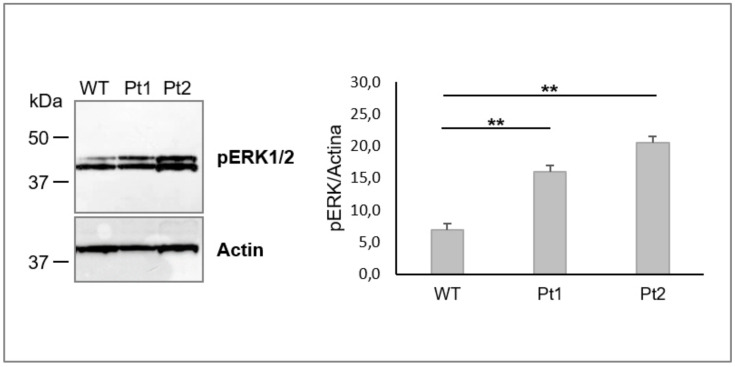
Analysis of phospho-extracellular-signal-regulated kinase 1/2 (pERK1/2) expression in U251 astrocytoma cells overexpressing MLC1 WT or carrying Pt1 and Pt2 pathological mutations. (**A**) WB analysis performed on 40 µg of protein cytoplasmic extract shows a significative inhibition of pERK1/2 in MLC1 WT cells when compared to the mutant Pt1 and Pt2 cells. Actin is used as a loading control. Molecular weight markers are indicated on the left (kDa). One representative experiment out of three performed is shown. (**B**) Densitometric analysis of pERK1/2 protein bands normalized with the amount of actin in the same samples is evaluated for each cell fraction. The bar graphs represent means ± SEM of three independent experiments. Statistical difference was calculated using the Kruskal-Wallis test (** *p* < 0.01).

**Figure 7 cells-09-01425-f007:**
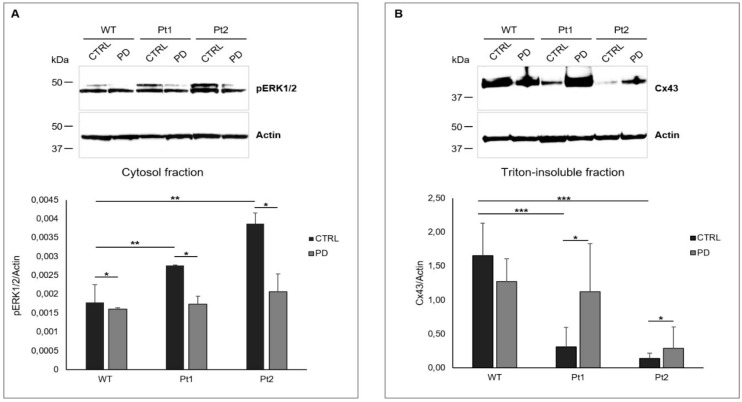
pERK1/2 inhibition rescues Cx43 partitioning in gap-junction fractions in U251 cells expressing MLC1 mutants. (**A**) WB analysis performed with 40 µg of cytoplasmic protein extract of MLC1 expressing cell lines, untreated (CTRL) or treated with 50 µM of the pERK1/2 inhibitor PD98059 (PD) for 1 h, confirms the inhibition of pERK1/2 activation in all the U251 cell lines analyzed. Actin is used as a loading control. Molecular weight markers are indicated on the left (kDa). One representative experiment out of three performed is shown. Densitometric analysis of pERK1/2 protein bands normalized with the amount of actin is evaluated for each cell fraction. The bar graphs represent means ± SEM of three independent experiments. Significant differences are calculated using the Kruskal-Wallis test followed by Dunn’s Multiple Comparison post hoc test. (* *p* < 0.05 and ** *p* < 0.01). (**B**) WB analysis of Triton-insoluble, gap-junction fractions (60 µg of protein extract) of MLC1-expressing cell lines, untreated (CTRL) or treated (PD) for 1 h with PD98059 (50 µM), reveals increased Cx43 levels in Pt1- and Pt2-treated cells when compared to their untreated controls (CTRL). No strong alterations of Cx43 distribution occurs in PD-treated WT MLC1 cells when compared to the corresponding untreated cells (CTRL). Actin is used as a loading control. Molecular weight markers are indicated on the left (kDa). One representative experiment out of five performed is shown. Densitometric analysis of the Cx43 protein band normalized with the amount of actin in the same samples is evaluated for each cell fraction. The bar graphs represent means ± SEM of five experiments. Statistical difference was calculated using the Kruskal–Wallis test followed by Dunn’s Multiple Comparison post hoc test. (* *p* < 0.05 and *** *p* < 0.001).

**Figure 8 cells-09-01425-f008:**
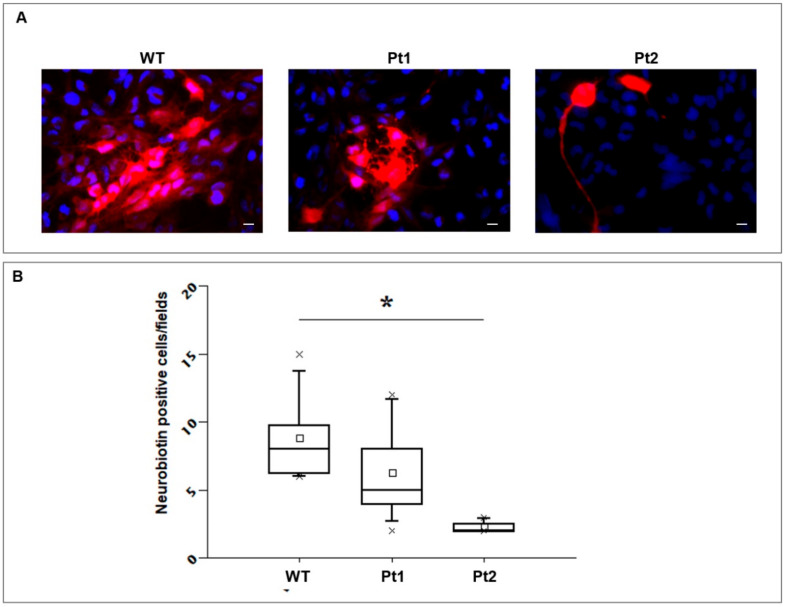
Evaluation of gap-junction permeability by the neurobiotin diffusion assay: The whole-cell configuration of the patch-clamp technique was used to inject the gap-junction permeable agent neurobiotin in single U251 astrocytoma cells in order to evaluate its differential diffusion in cell lines overexpressing MLC1 WT or carrying Pt1 or Pt2 pathological mutations. (**A**) After 10 min of neurobiotin injection, fixation, and incubation in streptavidin-TRIC/DAPI, fluorescence images were taken for further analysis. Exemplifying images of merged streptavidin-TRIC and DAPI fluorescence channels (red for neurobiotin and blue for DAPI) are shown in the panels. The number of neurobiotin-diffused cells is higher in WT MLC1 cells compared to cells expressing Pt1 and Pt2 pathological mutations, with the latter showing the lowest number of neurobiotin-positive cells. (**B**) The number of neurobiotin-positive cells in the U251 lines expressing MLC1 WT or carrying Pt1 or Pt2 pathological mutations were pooled and distributed as a boxplot. ANOVA analysis was used for testing statistical significance. The three groups were shown to be statistically different (*n* = 3–7; * *p* < 0.05), with MLC1 WT cells reporting the highest number and Pt2 cells reporting the lowest number of neurobiotin-diffused cells.

**Figure 9 cells-09-01425-f009:**
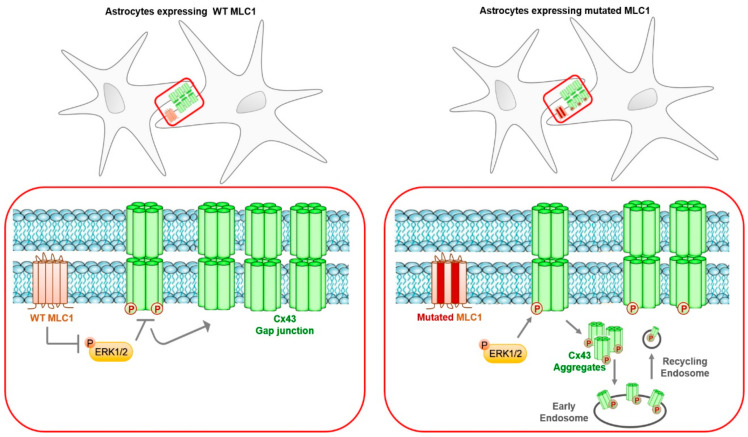
Schematic representation of the possible mechanism of the MLC1-mediated Cx43 regulation in astrocytes.

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
