# Peer review of "Megalencephalic Leukoencephalopathy with Subcortical Cysts Disease-Linked MLC1 Protein Favors Gap-Junction Intercellular Communication by Regulating Connexin 43 Trafficking in Astrocytes"

_cells, 2020, doi:10.3390/cells9061425_

Round 1

Reviewer 1 Report

The authors address a potentially interesting and important problem (the relationship between MLC1 and Cx43 in astrocytes) and its relation to disease. However, there are multiple problems with the paper:

  1. The authors need to define their abbreviations. They need to detail the full name of MLC1 (or several of the names) in the title and abstract. They need to write out the full name of the disease (megalencephalic leukoencephalopathy with subcortical cysts 1)
  2. The western blots and immunofluorescence need to include uninfected (parental) control cells. This would help to demonstrate the level of endogenous or background MLC1 expression. It would provide some evidence to support their contention of a “unspecific protein recognized by the anti-MLC1 pAb”. The authors need to present RNA data to prove that wild type and mutant mRNAs are expressed at similar levels.  No information is presented to support the claim that the mutants give rise to  “an early truncated and probably quickly degraded protein product.” The IF is unconvincing. What do the authors interpret as membrane staining? There is no evidence presented that MLC1 staining corresponds to ER.
  3. How did the authors prepare the “cytosol” fraction? Why is there any of the membrane protein Cx43 in this fraction? The experiment in Fig. 2 needs to show both the Triton soluble and insoluble fractions so that the amounts of Cx43 in total can be added up.
  4. 3. Appositional (junctional) staining should be quantified. Why are the nuclei stained? Colocalization of MLC1 and Cx43 should be enlarged and should be quantified.
  5. 4 co-localization should be quantified
  6. 5 needs to show treatment with EGF alone and CHX alone for WT and for each patient. Otherwise a meaningful comparison is not possible.
  7. The results section regarding use of the antibody recognizing the phosphorylated S262 residue should be deleted from the paper, since the authors detected no specific band “probably due to technical still unresolved issues”

Author Response

Answers to Reviewers

We thank the Editor and the Reviewers for their helpful suggestions and stimulating comments. All the new experiments and the modifications/corrections required by the Reviewers have been performed and the manuscript has been revised and changed accordingly.

All the changes made in the manuscript are highlighted in yellow.

We hope that the modified new version of the manuscript entitled “The Megalencephalic leukoencephalopathy with subcortical cysts disease-linked MLC1 protein favors gap-junction intercellular communication by regulating connexin 43 trafficking in astrocytes” by Lanciotti and collaborators, will be suitable for publication.

Point-by-point answers to each reviewer are reported below.

Reviewer 1:

  • The authors need to define their abbreviations. They need to detail the full name of MLC1 (or several of the names) in the title and abstract. They need to write out the full name of the disease (megalencephalic leukoencephalopathy with subcortical cysts 1).

The abbreviations have been defined in the title/abstract and in the other manuscript sections, as indicated by the Reviewer. Moreover, a list of all the abbreviations used in the manuscript has been included.

  • The western blots and immunofluorescence need to include uninfected (parental) control cells. This would help to demonstrate the level of endogenous or background MLC1 expression. It would provide some evidence to support their contention of a “unspecific protein recognized by the anti-MLC1 pAb”.

  • The western blot (WB) image shown in Figure 1 has been modified to include results obtained with protein extracts derived from U251 mock-infected cells (cells infected with an empty vector and used as control of the MLC1 expressing lines). No MLC1 specific signal has been detected in the cytosolic and membrane fractions of these cells, while a low molecular weight band (indicated with an asterisk in Fig. 1) has been observed in the membrane extracts from mock-infected cells as well as from all the MLC1 expressing cell lines. This latter result demonstrates that this band corresponds to a protein not specifically recognized by the anti-MLC1 polyclonal antibody (pAb).

In addition, we did not observe any MLC1 staining in mock-infected U251 cells by IF experiments using the anti-MLC1 pAb. These results have been now shown in Supplementary Fig. 2. The Material and Methods (MM) and Results section have been modified accordingly (Pag.4, Lines 111, 114-115; Pag.9, Lines 292-293, 310; Fig. 1 legend).

  • The authors need to present RNA data to prove that wild type and mutant mRNAs are expressed at similar levels. No information is presented to support the claim that the mutants give rise to “an early truncated and probably quickly degraded protein product.

  • As suggested by the reviewer, RT-PCR experiments to analyze MLC1 mRNA have been performed on mock-infected cells and MLC1 expressing cell lines and now shown in panel B of Fig. 1.

One microgram of total RNA has been reverse-transcribed for each cell sample and amplified using MLC1 specific primers, as described in the Material and Methods (MM) section. The c.178-10 t>a (Pt1 mutation) is a homozygous splice site mutation in intron 2, upstream of exon 3, that disrupts the sequence between intron 2 and exon 3, causing a complete skipping of exon 3 and the consequent reduction of MLC1 mRNA size. Due to the aberrant translation of the MLC1 I and II transmembrane domains (Petrini et al., 2013; Boor et al., 2006; Patrono et al., 2003), Pt1 protein length is thus reduced of about 30 amino acids.

The c.177+1 g>t homozygous mutation (Pt2 mutation) hits the splice-donor GT sequence of exon 2, resulting in a TT transition. The Human Splicing Finder (HSF) bioinformatics tool predicts a donor splice site break in intron 2, leading to incorrect splicing of exon 2 (Petrini et al., 2013, and reference therein) and possible (total or partial) intron 2 retention. As expected, we observed an increase of the MLC1 mRNA size (Fig. 1B). However, after the missing splice-donor sequence, the intron 2 DNA sequence contains a premature in-frame stop codon that most likely leads to a truncated and rapidly degraded MLC1 gene product (Petrini et al., 2013). WB and IF staining of Pt2 mutant expressing cells confirm these speculations (Fig. 1A, C).

  • About the comparison of MLC1 mRNA expression levels among the different cell lines, as expected, very low levels of MLC1 mRNA were observed in the mock-infected cells (Ø), representing the endogenous MLC1 mRNA expressed in the U251 cells, as previously reported (Ambrosini et al., 2008). Much higher levels of MLC1 mRNA were detected in cells expressing WT and Pt1 mutation, without quantitative differences between them and in accordance with the high efficiency of the cytomegalovirus (CMV) constitutive promoter controlling MLC1 expression in the retroviral vectors used to create the MLC1+ cell lines. On the contrary, lower levels of MLC1 mRNA were found in Pt2 cells, probably due to messenger instability caused by the splicing site mutation which generates an aberrant mRNA.

All these new data and the corresponding explanations have been added in the MM (Pag.4, Line108-114; Pag.5, Lines 127-132) and in the Results section of the manuscript (Pag.8, Lines 272-283; Pag 9, Lines 294-306 and Fig.1 legend).

  • The IF is unconvincing. What do the authors interpret as membrane staining? There is no evidence presented that MLC1 staining corresponds to ER.

  • We have previously described that in U251 cells MLC1 is mainly distributed at the plasma membrane and endoplasmic reticulum (ER) compartments (Lanciotti et al., 2012). To further support these observations and following reviewer 1 and 3 requests (see below), we have performed double immunostaining of WT MLC1 expressing cells using Abs recognizing MLC1 in combination with a membrane (CD44) and an ER (calnexin) specific Ab, quantitatively evaluating MLC1 distribution in both compartments by protein colocalization analysis. These data are now included in the new version of the manuscript in Supplementary Fig. 1 and mentioned in the Results section at Pag.9, Lines 307-308.

  • How did the authors prepare the “cytosol” fraction? Why is there any of the membrane protein Cx43 in this fraction? 

  • Cytosolic fractions have been obtained by the use of a solubilization procedure in presence of low ionic strength detergents and absence of Triton X-100 (as described in MM and in the new reference 34) and low-speed centrifugation that allows separating cytosol and small organelles (i.e. endosomes, lysosomes) from plasma and ER membranes. The presence of the membrane protein Cx43 in this fraction is mainly due to Cx43 localization in these intracellular compartments. The following solubilization of the remaining pellet in presence of 1% Triton X-100 allows extracting Triton-soluble membrane proteins. We have specified this procedure in the MM section (Pag.5 Lines.154-160). Moreover, as suggested by reviewer 3, in cultured cells gap junction formation occurs at a lower level than in tissue, justifying the greater presence of Cx43 in the cytosolic fraction when compared to that in the membranes. A specific sentence explaining this issue has been added at Pag.11-12, Lines 371-374.

  • The experiment in Fig. 2 needs to show both the Triton soluble and insoluble fractions so that the amounts of Cx43 in total can be added up.

  • The membrane fractions described in Fig. 2 and mentioned through the whole paper represent the Triton-soluble fractions obtained by membrane solubilization in presence of 1% Triton X-100. To clarify this point, we specified it in the MM (Pag.5, Lines.154-160) and Results section (Pag.10, Lines 337-340) and inserted a new reference to clarify further technical details (ref.34). We also modified the title and legend of Panel B of Fig. 2, accordingly.

  • Appositional (junctional) staining should be quantified. Why are the nuclei stained? Colocalization of MLC1 and Cx43 should be enlarged and should be quantified. Co-localization should be quantified

  • As suggested by the Reviewer, we modified Fig. 3 including the enlargement of panel C image and performing quantification analysis of MLC1/Cx43 colocalization. The unspecific staining of U251 cell nuclei by the anti-Cx43 pAb has been already observed in U251 cells (reference 38 mentioned in the text at Page.11, Lines 371 ), and it could be due to cell-specific cross-reaction of the Cx43 pAb with some nuclear membrane proteins. All these new modifications have been inserted in MM (Pag.8, Lines 257-261) and un the Results section (Pag.11, Lines 367-369; Figure 3 legend).

  • They needs to show treatment with EGF alone and CHX alone for WT and for each patient. Otherwise a meaningful comparison is not possible.

  • Following Reviewer suggestion, new experiments have been performed to monitor, independently, the effects of cycloheximide (CHX) and epidermal growth factor (EGF) treatment on Cx43 expression/degradation kinetics in MLC1+ U251 cell lines. The results of these experiments indicated no significant differences between control and mutant MLC1 expressing cells. A new supplementary figure showing these results has been added (Fig. SD4). Accordingly, also the MM and Result section has been modified to include data description (Pag. 4, Lines 121-124; Pag.13-14, Lines 430-432).

  • The results section regarding use of the antibody recognizing the phosphorylated S262 residue should be deleted from the paper, since the authors detected no specific band “probably due to technical still unresolved issues”

  • As suggested by the Reviewer this part of the results has been deleted.

Reviewer 2 Report

For Authors

This study showed that the novel interaction between MLC1 and Cx43 in astrocyte, and provided new knowledge on MLC pathogenesis.

The originality and scientific soundness of this study are high, and experimental design is reasonable. This study can be expected to stimulate active discussion among a broad range of readers of Cells. I expect to develop to in vivo experiments in further.

Just minor points:

  1. Line 485 [cells]  c is italic
  2. Line 562 [of cells expressing the]  The font size of is small.
  3. Line 619 [by the EtBr]   maybe space after [by] is more.

Author Response

Manuscript cells-784463

Answers to Reviewers

We thank the Editor and the Reviewers for their helpful suggestions and stimulating comments. All the new experiments and the modifications/corrections required by the Reviewers have been performed and the manuscript has been revised and changed accordingly.

All the changes made on the manuscript are highlighted in yellow.

We hope that the modified new version of the manuscript entitled “The Megalencephalic leukoencephalopathy with subcortical cysts disease-linked MLC1 protein favors gap-junction intercellular communication by regulating connexin 43 trafficking in astrocytes” by Lanciotti and collaborators, will be suitable for publication.

Point-by-point answers to each reviewer are reported below.

  • Reviewer 2:

  • Extensive editing of English language and style required.

  • The manuscript has been revised for the English language and style. All minor points indicate below have been corrected.

  • Line 485 [cells] c is italic

  • The “c” in italic has been corrected

  • Line 562 [of cells expressing the]  The font size of is small

  • The font size has been corrected

  • Line 619 [by the EtBr]  maybe space after [by] is more

  • The indicated space has been corrected.

Reviewer 3 Report

The paper analyze the relationships between the gap junction protein Cx43 and MLC1, the astrocytic protein whose mutations are responsible for Megalencephalic leukoencephalopathy, and, therefore, it is an important contribution to the elucidation of the pathophysiological mechanism of this condition. The experiments were well designed and the manuscript is well written. It needs a few adjustments to be able to give more basement to the results obtained.

Minor review

In Introduction:

Lines 47 and 48 – it is not appropriate to quote the study in question as one reference

In Results:

Line 279 - The legend of Figure 1 is incomplete

Figures 1 and SD2 - Regarding the results obtained in the WB with the MLC1 antibody it is important to explain the presence of the two bands with different MW (37kDa and 60kDa)! Are they dimers? Why don't they appear in all lysates?

Generally, cultures cell lines are more difficult to form the gap junctions formed with connexins, which may justify the fact that greater expression occurs of cx43 in the soluble cytosolic fraction, as well the as few points where cx43 is located in areas of the gap junctions.

It is important to report which epitope of protein that he anti-CX43 recognize. The C-term or all sequence? In addition, it would mark of the protein degraded.

In discussion:

The paragraphs are too long, which hinders a better understanding of the text. They can be shorter and arranged in the same order as the results was described.

Major review

Are there were validated antibodies used to ensure their specificity in the experiments carried?

It would be adequate and contribute to the results obtained in this study if the authors performed IF with double labeling: anti MCL1 or anti IF-Xpress with specific antibodies for the endoplasmic reticulum and for the cytoplasmic membrane, to demonstrate the location with MCL1 protein. In addition, it can also be quantified the co-location this proteins by means of statistical analysis, e.g. the Pearson test, the Zeiss program itself makes these analysis available.

Author Response

Manuscript cells-784463

Answers to Reviewers

We thank the Editor and the Reviewers for their helpful suggestions and stimulating comments. All the new experiments and the modifications/corrections required by the Reviewers have been performed and the manuscript has been revised and changed accordingly.

All the changes made in the manuscript are highlighted in yellow.

We hope that the modified new version of the manuscript entitled “The Megalencephalic leukoencephalopathy with subcortical cysts disease-linked MLC1 protein favors gap-junction intercellular communication by regulating connexin 43 trafficking in astrocytes” by Lanciotti and collaborators, will be suitable for publication.

Point-by-point answers to each reviewer are reported below.

  • Reviewer 3:

  • In Introduction: Lines 47 and 48 – it is not appropriate to quote the study in question as one reference
  •  As suggested by the reviewer new appropriate references have been added (Ref. 3 and 6).

  • In Results: Line 279 - The legend of Figure 1 is incomplete

  • The complete legend of Fig. 1 has been inserted.

  • Figures 1 and SD2 - Regarding the results obtained in the WB with the MLC1 antibody it is important to explain the presence of the two bands with different MW (37kDa and 60kDa)! Are they dimers? Why don't they appear in all lysates?

  • We have previously demonstrated that in astrocytes the MLC1 protein can be found as monomer (of about 30-36 kDa of MW), mainly localized in the cytosolic but also in the membrane compartments, and as dimer (of about 60-64 MW) that is exclusively localized in the membrane extracts (Ambrosini et al., 2008, Brignone et al., 2011). A sentence explaining these features has been added to the Results section (Pag.8, Lines 283-287) and Fig.1 legend. We previously described that pathological mutations can affect total protein amount and its localization at plasma membranes when expressed in astrocytoma cells (Lanciotti et al., 2012). We also found patient mutations affecting dimer formation (manuscript in preparation) such as the two, newly cloned mutations, described in the present paper.

  • Generally, cultures cell lines are more difficult to form the gap junctions formed with connexins, which may justify the fact that greater expression occurs of cx43 in the soluble cytosolic fraction, as well the as few points where cx43 is located in areas of the gap junctions.

  • We completely agree with the Reviewer observation. We highlighted this concept in a specific sentence in the Results section (Pag. 11-12, Line 371-374).

  • It is important to report which epitope of protein that the anti-CX43 recognize. The C-term or all sequence? In addition, it would mark of the protein degraded. 

  • The anti-human Cx43 polyclonal (p)Ab used in the present work is raised against the C terminal peptide (362-382 aa, KPSSRASSRASSRPRPDDLEI) of the Cx43 protein, as now indicated in the MM (Pag.5, Line.139). The molecular weight of the protein we detected by WB corresponds to the whole protein. We rarely observed some lower molecular weight bands by WB, probably representing protein degradation products, in all the cell lines analyzed.

  • In discussion: The paragraphs are too long, which hinders a better understanding of the text. They can be shorter and arranged in the same order as the results was described.

  • The discussion has been modified and shortened as suggested by the Reviewer.

  • Are there were validated antibodies used to ensure their specificity in the experiments carried?

  • The anti-MLC1 pAb we used for WB and some IF staining has been generated using the whole human recombinant protein and extensively used for WB and IF of astrocytes in in vivo and in vitro experiments (Ambrosini et al., 2008; Brignone et al., 2011; Lanciotti et al., 2012, 2016). No unspecific staining was observed in human and rat/mouse brain tissue where MLC is detected only in perivascular astrocytes and no other cellular populations (Ambrosini et al., 2008), confirming the specificity of the Ab generated.

The other Abs used in the present work are commercially Abs extensively used to specifically detect the corresponding proteins, as indicated in each manufacturer’s datasheet.

  • It would be adequate and contribute to the results obtained in this study if the authors performed IF with double labeling: anti MCL1 or anti IF-Xpress with specific antibodies for the endoplasmic reticulum and for the cytoplasmic membrane, to demonstrate the location with MCL1 protein. In addition, it can also be quantified the co-location this proteins by means of statistical analysis, e.g. the Pearson test, the Zeiss program itself makes these analysis available. 

  • As suggested by the Reviewer colocalization experiments and quantification by statistical analysis have been performed and included in the new Supplementary Fig.1 and described in the Results Section Pag.9, Lines.306-308.

Round 2

Reviewer 1 Report

My original criticisms have been adequately addressed